# Dog Walking before and during the COVID-19 Pandemic Lockdown: Experiences of UK Dog Owners

**DOI:** 10.3390/ijerph18126315

**Published:** 2021-06-10

**Authors:** Sara C. Owczarczak-Garstecka, Taryn M. Graham, Debra C. Archer, Carri Westgarth

**Affiliations:** 1Department of Livestock and One Health, Faculty of Health and Life Sciences, Institute of Infection Veterinary and Ecological Sciences, University of Liverpool, Leahurst, Chester High Road, Neston CH64 7TE, UK; owczarcz@liverpool.ac.uk (S.C.O.-G.); taryn.mary.graham@liverpool.ac.uk (T.M.G.); 2Department of Equine Clinical Studies, Faculty of Health and Life Sciences, Institute of Infection Veterinary and Ecological Sciences, University of Liverpool, Leahurst, Chester High Road, Neston CH64 7TE, UK; darcher@liverpool.ac.uk

**Keywords:** COVID-19, pandemic, dog walking, dog ownership, lockdown, public health

## Abstract

Background: This study investigated the impacts of the first COVID-19 UK lockdown on dog walking and ownership. Methods: An online survey was circulated via social media (May–June 2020). Completed responses (n = 584) were analysed using within- and between-group comparisons, and multivariable linear and logistic regression models were created. Open-ended data were coded into key themes. Results: During lockdown, dogs were walked less frequently, yet for a similar duration per week and closer to home. Dogs whose owners lived alone, or whose owners or household members had heightened vulnerability to COVID-19 were walked less than before, as were high-energy dogs. A minority of owners continued dog walking despite exhibiting symptoms or needing to self-isolate, justifying lack of help, dog behavioural problems, living in less populated areas, and the importance of outdoor exercise for their mental health. Dog ownership had multiple benefits (companionship, purpose and motivation; break from bad; positive to focus on) as well as challenges (changes in dog behaviour, balancing dog needs with public health guidance, accessing pet food/supplies and services, and sharing crowded outdoor spaces with others). Most did not have an emergency care plan for their pet before the pandemic and only a handful developed one. Conclusions: Findings can be used to inform public health and dog welfare strategies for future lockdown situations or other disasters and emergencies likely to impact on daily routines.

## 1. Introduction

Dog walking is an important public health practice that contributes to human health and dog welfare [1,2]. A recent UK study demonstrated that dog owners were four times more likely to meet the minimum physical activity guidance of 150 min per week than their non-dog-owning counterparts, substantially higher than found in other countries [3]. In addition to motivating owner exercise, dog walking has mental health benefits [4], particularly the longer ‘recreational walks’ undertaken after work and on weekends [5]. Exercise levels are also an important predictor of canine obesity [6], and amounts given vary widely within and between dog breeds [7]. A UK study estimated that on average, dogs are exercised 7 times per week, for a total of 220 min per week (approximately 30 min per walk) [3]. Once daily walking is considered a minimum requirement of a responsible dog owner [8], but frequency and duration of dog walks are affected by a number of factors, including dog size, access to suitable public spaces for dog walking, and the nature of the relationship between the dog and their owner(s) [9].

Strict ‘first lockdown’ measures (thereafter, ‘lockdown’) were imposed in the UK from 23 March 2020 until 10th May to control COVID-19 transmission between people. Specific rules between the UK nations varied, but lockdown restrictions allowed people to only leave their homes for medical help, to shop for essential items (e.g., food or medicines), perform essential work, and for once-a-day outdoor exercise [10,11]. Exercise was further restricted to local areas and for up to 1 h in some regions [10,11]. It was unclear whether these restrictions applied to dog walking [12], resulting in potential confusion and variability within and between regions. Dog walking was not explicitly addressed in English guidelines [13], and was interpreted in different ways by local authorities [11]. Scottish guidelines meanwhile stated that dogs could be taken out more than once a day [14], whereas guidelines for Northern Ireland required owners to include dog walking within their once-a-day exercise [15]. These rules and changes to people’s routines due to working from home or being furloughed [10] are likely to have altered the frequency, duration, and locations that owners walked their dogs and their experiences during dog walking. 

Pandemic-related lockdowns are known to have impacted on pet owners and pets themselves in many countries. Data, including information obtained from the UK, suggests that dogs were identified as being a source of purpose, routine, sanity and entertainment for people during lockdown [16,17,18,19] and having a pet was protective against reduced mental health and increased loneliness [19]. An Australian study of people living alone also found that dog owners reported feeling less lonely during lockdown than people without a pet [20]. Having an excuse to leave the house and do exercise was an important aspect of owning a dog at this time, resulting in increased social interactions [17,20], with some giving their dogs more exercise than normal [16,20]. Likewise, in Canada, family dog ownership was positively associated with more active behaviour during COVID-19 restrictions, even as children had overall decreased physical activity, spent less time outside, and had increased sedentary behaviour and sleep [21]. 

Despite these overall benefits, owners expressed a number of concerns about dog walking during lockdown and around reduced quality of life for their pets. This included consequences related to closure of dog parks during lockdown and dogs not being allowed to exercise off-leash in some locations [22,23]. In the UK, dogs were more likely to have just one walk per day instead of 2–3, and whilst the total duration of daily walking exercise remained similar due to longer walks, dogs were less likely to be walked off-leash, were walked in less rural areas, and were less likely to be allowed to meet with other dogs [16,24]. Some owners considered that walks had become less enjoyable, because of perceived new dog owners in their area who did not respect social distancing or whose dogs were allegedly out of control [16,17]. UK pet owners also reported concerns about the impact of lack of socialisation, especially for puppies [17,24]. 

Others worried about contracting COVID-19 during dog walking. At the start of the pandemic, concerns about risk to human health posed by potential cross-species transmission of COVID-19 by dogs was thought to be low with human-to-dog transmission more likely than dog-to-human transmission [25]. A Spanish survey identified that 7% of respondents had concerns that dog walking increased the risk of COVID-19 infection [23], findings echoed by respondents to a UK-based survey [17]. Another Spanish study suggested that people who walked dogs were 78% more likely to report contracting COVID-19 [26], due to dogs being infected themselves or being a source of physical introduction of the virus into the house [26]. However, it is known from previous research that social interaction with other people is a common outcome of dog walking [4], and thus, increased reported chances of infection may have been due to physical proximity to other people (e.g., stopping and talking to each other), rather than virus transmission associated with the dogs themselves. 

To date, studies focused on the impact of COVID-19 on dog ownership have explored various facets of human–dog interactions, including walking practices. However, these studies did not distinguish between walking undertaken by the dog, and a person’s activity walking with their dog, which may be different, particularly for multi-person households. The first aim of this study was to compare dog walking behaviour of UK owners and their dogs before and during the first lockdown (March/April 2020). We hypothesised that there may be differences in dog walking participation in single-person households compared to households with multiple members who could share the responsibility. Given the unprecedented changes in lifestyle triggered by the pandemic, the second aim was to invite dog owners to openly share the challenges and benefits of keeping and caring for a dog during these early stages of the pandemic.

## 2. Materials and Methods

### 2.1. Study Design and Methodology

An online survey was conducted in the UK between 24 May and 11 June 2020. This study was approved by the University of Liverpool Veterinary Research Ethics Committee (VREC957) and was distributed via social media, including Facebook and Twitter. Many people acquired new pets during lockdown (e.g., “pandemic puppies” or rescue adoption [27]); however, our study recruited only those who had their dog both before and during this time, so that changes could be assessed. Dog owners also had to live in the UK and be over 18 years of age to be eligible to participate. 

### 2.2. Survey Design

The questionnaire consisted of 64 open- and close-ended questions divided into 8 sections. Possible response options for questions not used in further analysis are listed below; the remaining options are provided in Appendix A: •“About my dog”: number of dogs in the household; their size; age; sex; owner’s subjective assessment of their dog’s perceived energy levels (response categories were: low energy, medium energy and high energy; no further definitions or descriptions of behaviours that exemplify these categories were provided); and owner-perceived relationship with their dog (based on the Inclusion of the Other in the Self question adapted for pets [28,29]). Respondents were presented with seven images of circles with a progressing degree of an overlap between a circle representing a dog and an owner: starting with completely separate circles, representing a weak relationship, and ending with two nearly completely overlapping circles, representing a strong relationship). For questions relating to only one dog, respondents were asked to answer based on the dog they felt emotionally closest to. This strategy was chosen because dog walking is known to be associated with the strength of the relationship between the owner and the dog [9], and therefore this was the dog most likely to be walked pre-lockdown and thus affected by it.•“Walking this dog”: frequency of dog’s interactions with people and other dogs on walks before the lockdown; dog’s perceived recall reliability (response categories: dog never comes back when called, rarely, sometimes, often); weekly frequency and duration (in minutes) of dog walking undertaken by dog(s) during the lockdown and before (based upon the Dogs And Physical Activity tool [30]); perceived changes in total number of walks a dog gets from anyone since the lockdown; and perceived changes in how often a dog is off-lead, allowed to interact with other dogs and people since the social distancing measures were in place.•“Who walks dogs”: who walked the dog during lockdown and before; whether the person who walks the dog changed since the lockdown; and if so, how (open-ended question).•“Personal dog walking”: weekly frequency and duration (in minutes) of dog walking undertaken by the respondent during lockdown and before [30]; perceived changes in respondent’s number of dog walks since the lockdown and daily number of steps taken pre and during the first national lockdown (the type of a recording device was not specified, respondents may have used mobile phones or smart watches).•“Other dog walking”: location of dog walking both before and since the lockdown; perceived changes to walking location; description of these changes (open-ended question), whether COVID-19 changes brought the respondents into greater contact with livestock when walking; and attitude-related questions regarding experience of dog walking during the lockdown (e.g., Going for a dog walk offers a break from COVID anxiety). Responses to these questions were presented on a 5-point Likert scale anchored with “strongly agree” and “strongly disagree”.•“Perceptions of dog ownership”: whether caring for a dog during the period of social isolation has been challenging and helpful; two open-ended questions enquiring about ways in which caring for a dog was challenging and helpful during the lockdown; presence of an emergency care plan (defined in the survey as “verbal or written agreement about who would care for your dog if you were ill or other plans for an emergency”) and whether the respondent made one since the coronavirus outbreak.•“COVID questions”: whether the respondent experienced suspected COVID-19 disease and if so, whether they and household members walked their dog during the period when they had symptoms and when symptoms were not present any more but they were still within the designated isolation period; whether the respondent or household member was vulnerable and told to isolate for 12 weeks regardless of the symptoms and whether the vulnerable person continued to walk the dog whilst isolating.•“About you”: respondent’s age, gender, qualifications, and an open-ended questions about anything else regarding experience of dog walking during the coronavirus outbreak.

Some closed-ended questions included answer choice ‘Other- please specify’. These answers were re-coded by one of the co-authors (CW) where possible to match the pre-specified answers (e.g., in local woods was re-coded into pre-existing category “Woodlands”), or a new category was created (e.g., in response to the question about where dogs were walked during the lockdown, a category “private field” was created). 

### 2.3. Data Handling and Statistical Analysis

#### 2.3.1. Data Handling

As part of checking the data for imputation errors, in line with previously published cut-off points, weekly walking duration was capped at 2520 min maximum (approximately 6 h a day) [31]. Values greater than this were considered as entered in error and were converted to 2520 ahead of the analysis. People living alone were identified by filtering those who answered “I live alone” to the question about other household members experiencing COVID-19 symptoms and the question about other household members being classed as vulnerable. Contradictory answers (e.g., answering “I live alone” to the first question and “Yes [other household members are vulnerable]” to the second one) were not used in the analysis. Relationship with a dog was recoded into 3 categories: strong (combined answers 6 and 7), medium (combined answers 5 and 4) and weak (combined 1, 2 and 3). All analysis was conducted in R (4.0.3) [32]. 

#### 2.3.2. Multivariable Regression Analysis

To characterise our sample within the context of previous research on dog walking, multivariable generalised models with dog’s and person’s walking durations pre-lockdown used as an outcome variable were constructed. In both models, the outcome variable was log transformed. Dog-related variables (size, age, sex, energy levels, relationship with the owner), owner-related variables (age, gender, education, living arrangements: living alone or with others) and household-related variables (number of dogs in the household) were used as predictor variables [9]. Model fit is reported by describing F value, degrees of freedom, Cragg–Uhler Pseudo R^2^, and Akaike Information Criteria (AIC).

#### 2.3.3. Within- and between-Group Comparisons 

Data distributions were explored visually and with Shapiro–Wilks tests. As dog-walking durations were not normally distributed, the paired within-group changes (before-during lockdown) in weekly walking duration per dog (dog’s walking duration), owner (person’s walking duration) and in the number of steps taken as recorded by personal tracking devices were explored with the Mann–Whitney sum rank paired-samples test for comparison of distributions [33]. Changes in who walked the dog and locations where dogs were walked were explored with Cochran’s Q test extension of McNemar–Bowker tests. Comparison of where a dog was walked and presence/absence of an emergency care plan between those who did and did not live alone was carried out with a chi-square test. Analysis of differences in data distribution for dog’s and person’s walking was conducted for both the frequency and duration of dog walking, before and during the lockdown. For variables with multiple categories (i.e., living arrangements, number of dogs in the household, dog size, dog age, dog’s energy, relationship with a dog) data were analysed with Mann–Whitney test for unpaired data (where there were 2 categories) or Kruskal–Wallis (K-W) test (where there were more than 2 categories). Where the K-W test was significant, post-hoc follow-up Dunn tests with Benjamini–Hochberg (B-H) correction for multiple comparisons were used [34]. 

#### 2.3.4. Logistic Regression Analysis 

Two multivariable logistic regression models were created to explore (1) dog’s and (2) person’s walking changes during the lockdown (reduced total walking duration per week, compared to combined category “stayed the same or increased”). A full model was created using owners’ and household COVID-19 symptoms, owner and household self-declared heightened vulnerability to the virus, having an emergency care plan for the dog and owner- and dog-related demographic variables as predictor variables. Backward elimination method was used to identify significant variables (*p* < 0.05).

### 2.4. Qualitative Data Analysis

Thematic analysis was carried out. Answers to open-ended questions were coded line-by-line by the co-authors (CW and TMG), by assigning codes that captured the main sentiments conveyed within and across responses [35]. Initial codes were revised throughout the coding process to ensure codes accurately reflected the text. Codes were then compared and grouped and patterns emerging from the initial coding were discussed between the co-authors. After further revision the main themes were discussed and identified [35].

## 3. Results

### 3.1. Owner and Dog Characteristics 

A total of 995 respondents began/half-completed the survey. Of these, 941 were eligible, and 584 completed the survey to the end and were included in the analysis. 

Most respondents had 1 dog (56.0%, n = 327), followed by 2 dogs (29.8%, n = 174) and 3 or more (14.2%, n = 83). Most dogs were 1–5 years of age (50.7%, n = 296), 35.8% (n = 209) were aged 6–10, 8.6% (n = 50) were age 11+ and 5.0% (n = 29) were under 1 year of age. Slightly over half of dogs were male (53.4%, n = 312). Most dogs were described as medium size (41.7%, n = 242), 35.0% (n = 203) as large/giant and 23.3% (n = 135) as small/toy sized. Over half of dogs were considered medium energy (55.5%, n = 324), 7.2% (n = 42) as low and 37.3% (n = 218) as high energy. 

Owners were mostly 30–50 years of age (43.2%, n = 252), followed by over 50 (37.9%, n = 221) and 18–29 (18.4%, n = 107). The majority identified as female (88.7%, n = 516), 10.5% (n = 61) identified as male and gender was not disclosed for 0.9% (n = 5). Most held a university level degree (69.0%, n = 400). Six respondents provided contradictory answers to the question to which “I live alone” was one of the answers. These responses were removed and 15.5% (n = 89) respondents were identified as living alone. Most respondents (91.4%, n = 533) had not developed COVID-19 symptoms at the time of questionnaire completion. However, 16.3% (n = 95) reported that someone else in their household had experienced symptoms. A small proportion of respondents (12.4%, n = 72) were classified as or considered themselves as vulnerable and were expected to isolate for 12 weeks. 

Prior to the lockdown, most respondents (62.3%, n = 363) did not have an emergency care plan for their dog. Of those who did not have a care plan, 7.7% (n = 28) prepared one during the pandemic. Before the lockdown, a significantly higher proportion of those living alone (57.3%, n = 51) had a care plan compared to those living with other household members (33.5%, n = 163, *p* < 0.001). 

### 3.2. Dog’s Walking before Lockdown

A multivariable regression model showed that longer walk duration before lockdown was associated with the following dog-related characteristics: large/giant dog size (compared to small/toy size) (*p* = 0.01) and high energy levels (compared to perceived low energy levels) (*p* < 0.001). In the same model, owner-related characteristics associated with longer walk duration included owners aged 30–50 years old (*p* = 0.001) and over 50 years of age (*p* = 0.03) compared to 18–30 years (Appendix A). Owners education to a A-level or equivalent level was negatively associated with dog walking duration compared to those educated to degree level (*p* = 0.01, Appendix A). 

### 3.3. Differences in Dog’s Walking before and during Lockdown

Most respondents stated that there had been no difference in the weekly frequency of dog’s walks (from anyone) during lockdown (52.8%, n = 306), 27.8% (n = 161) reported a decrease and 19.5% (n = 113) reported an increase. Overall, dogs experienced a significant decrease in the frequency of walks during lockdown, with a weekly median frequency of walks for dogs reduced from 10 to 7 (*p* < 0.001 Table 1). Weekly frequency of dog’s walks during lockdown compared to before was significantly reduced for: dogs of owners living alone (*p* = 0.04) and those living with others (*p* = 0.009); single-dog (*p* = 0.03) and two-dog households (*p* = 0.04); dogs of medium (*p* = 0.04) and large/giant-size (*p* = 0.013); dogs aged 6–10 years old (*p* = 0.012); dogs perceived to be of high (*p* = 0.023) and medium energy (*p* = 0.04); and where owners rated the relationship strength as medium (*p* = 0.04) or strong (*p* = 0.02; Table 1). In contrast to before lockdown, there was evidence that dogs were walked more frequently during lockdown when from single- compared to two-dog households (*p* = 0.029), but no other significant differences were observed (Table 2). Total duration of walks before lockdown was significantly less in dogs perceived to have low compared to medium (*p* < 0.001) or high (*p* < 0.001) energy levels (Table 2).

Comparison of dog’s walking duration before and during the pandemic lockdown revealed no overall change (*p* = 0.41; median of 420 min, Appendix A). Similar to the pattern observed for frequency of walks, before lockdown, dogs rated as low energy were walked for a significantly shorter weekly duration than those rated as high (*p* < 0.001) or medium (*p* = 0.04) energy (Table 3). A logistic regression model (Table 4), demonstrated that a reduction in dog’s total weekly walking duration during the lockdown was associated with dog owners who were vulnerable or were living with a vulnerable household member compared to not (*p* = 0.04) and nearly significant with owners who lived alone compared to with others (*p* = 0.06). The results of the multivariable logistic regression before backward elimination are summarised in the Appendix A.

Only 9.4% (n = 55) of dogs had been brought into closer contact with livestock compared to before the lockdown. Further changes in social interactions on walks and how dogs were managed are summarised in Figure 1.

### 3.4. Person’s Dog Walking before Lockdown

Prior to lockdown, between group comparison suggested that a person’s dog walking per week was significantly more frequent where they rated the relationship with the dog as strong (*p* = 0.012) or medium (*p* = 0.029) compared to weak, and strong compared to medium (*p* = 0.021, Table 2). Greater weekly dog walking duration was significantly associated with living alone (*p* < 0.001), dogs with high/medium energy levels compared to low (*p* < 0.001), and rating the relationship with a dog as strong or medium compared to weak (*p* < 0.05, Table 3). Multivariable regression modelling identified that an increased total weekly dog walking duration of a person was associated with having 3 or more dogs in the household (*p* = 0.03), dog’s high energy levels (*p* = 0.02), owners’ age 30–50 (*p* < 0.02), and owners living alone (*p* = 0.01). A reduced walking duration was associated with owners being educated at an A-level or equivalent level (*p* < 0.001, Appendix A).

### 3.5. Differences in Person’s Dog Walking before and during Lockdown

When asked directly about perceived change in person’s dog walking, 48.8% of respondents (n = 273) reported no change, 27.0% (n = 151) reported a decrease and 23.6% (n = 132) reported an increase in the total number of weekly walks. During lockdown, within-group comparison demonstrated that reduced frequency was again significantly associated with a weaker relationship with the dog (Table 1). Frequency of person’s dog walking was significantly reduced during lockdown for people living alone (*p* = 0.02) and with others (*p* = 0.03), for two-dog households (*p* = 0.009), large/giant dogs (*p* = 0.04), dogs aged 6–10 years (*p* = 0.006), dogs rated to have high energy levels (*p* = 0.05) and where owners rated the relationship with dogs as strong (*p* = 0.01) (Table 1). 

No significant differences in person’s walking duration were observed overall and the total median weekly durations during the lockdown compared to before were similar (Table 3 and Appendix A). Similar to pre-lockdown, person’s walking duration during the lockdown was significantly longer when living alone compared to living with others (*p* < 0.001) and where dog energy levels were rated as high or medium compared to low (*p* < 0.001) (Table 3). A logistic regression model (Table 5), demonstrated that a reduction in person’s total weekly walking duration during the lockdown was associated with owners who lived alone compared to with others (*p* = 0.02). The results of multivariable logistic regression analysis before backward elimination are summarised in the Appendix A. 

Among the owners who provided a daily step count based on personal activity trackers (n = 106, 18.2% of all respondents), the mean number of steps taken daily before the lockdown (with and without dogs) was 9996 (SD = 3483.28) before and 9147.2 (SD = 3640.24) during lockdown. The median change of averaged daily step count was 568.6 fewer steps during the lockdown, but this change was not significant (*p* = 0.14). 

Nearly one-third of respondents (28.8%, n = 168) stated that a different person was walking their dog compared to pre-lockdown. Compared to before, during lockdown, significantly fewer people used dog walkers (*p* < 0.001) or other friends/family (*p* < 0.001) and more walked the dog themselves (*p* < 0.001) (Figure 2). Qualitative analysis of open-text responses indicated that owners stopped using professional dog walking/care services or informal friends and family, because they were not allowed or because they did not need it any more as somebody else was able to walk the dog (working/studying from home, or being on furlough). However, others had no choice but to rely on friends and family for help due to a loss of professional dog walkers. Lockdown sometimes led to household members spending more time walking dogs together than previously, but more commonly they shared duties. Some household members and other carers who would normally dog walk were not able to due to COVID-19 symptoms or shielding.

### 3.6. Change in Walking Location 

During lockdown, walking on streets (*p* < 0.001) increased, but decreased for urban parks (*p* < 0.001), beaches (*p* < 0.001), woodlands (*p* < 0.001), and country parks (*p* < 0.001) (Figure 3). Qualitative analysis of open-text responses indicated that as driving had been limited to essential travel only during lockdown, and concerns that some places and their car parks had closed, dog walking “stayed local,” with many happening directly “from the front door only”, “round the block/streets [and] then home”, often “rediscovering local footpaths”. Summary of questions on experience of dog walking during the pandemic lockdown, including questions regarding travel for walking is summarised in Figure 4. Most respondents stated that they were travelling shorter distances to dog walking locations 62.8% (n = 364) and agreed or strongly agreed with the statement that they had access to sufficient local areas within walking distance from their house to walk their dog (83.6%, n = 483). Significantly more of those living alone changed their walking location compared to those who did not live alone (*p* < 0.001). 

### 3.7. Dog Walking and Attitudes to COVID-19

Of those who experienced COVID-19 symptoms (n = 95), 38% (n = 19) continued to walk their dog when experiencing symptoms and 34% (n = 17) walked the dog when symptoms had passed but when they were still within the designated isolation period. A minority (3.2%, n = 3) of respondents reported that other household members were walking the dog when showing COVID-19 symptoms and 6.3% (n = 6) within the designated isolation period. Qualitative analysis highlighted that reasons for continuing to walk included not having anyone else to help; not trusting anyone else to help because dog is reactive (i.e., responds by pulling, lunging, barking, or growling to other dogs/people); living in rural areas or having access to own private land where they never see other people; presented symptoms early on in February, when little was known about the virus; and believing it was important for mental health, so long as extra precautions were taken (e.g., walking early in the morning or late at night to avoid others). 

### 3.8. Benefits of Caring for a Dog during Lockdown

The majority of respondents reported that owning/caring for a dog during lockdown had been beneficial 91.2% (n = 529). Further summary of attitudes and experiences of walking during the lockdown and its impact on mental health are provided in Figure 4. The key themes identified in the course of qualitative analysis as benefits of caring for a dog during lockdown were: Companionship; Purpose and motivation; A break from the bad; A positive focus (Table 6). 

### 3.9. Challenges of Caring for a Dog during Lockdown

Owning and caring for a dog during the period of isolation/distancing had been challenging for a small proportion of respondents 16.3% (n = 95). Further summary of attitudes and challenging experiences related to dog walking during the pandemic lockdown is shown in Figure 4. The qualitative analysis elucidates the key themes identified as challenges for dog ownership and walking during lockdown as: concerns about actual or anticipated dog behavioural impacts; balancing public health guidelines with meeting dog’s welfare needs; conflicts in using crowded outdoor spaces with other users and coping strategies; risk of contracting COVID-19; accessing pet food, supplies and pet services (Table 7). 

## 4. Discussion

This is the first study to explore changes in dog walking of both owners and their dogs in the UK during the first lockdown (March/April 2020) of the COVID-19 pandemic. Government restrictions implemented to manage the transmission of COVID-19 and to prevent economic difficulties, including working from home and furlough schemes [10], are likely to have altered the frequency and locations that owners walked their dogs and their experiences during dog walking. Overall, results suggest good general compliance with emergency government regulations including being restricted to once daily outdoor exercise and reduced contact with other people. Dogs may have been walked less often per week, yet total weekly duration of walks stayed the same, meaning dog owners stayed out longer when they did walk. Compared to pre-lockdown, people walked their dog(s) closer to home. A potential public health risk was posed by a minority of owners or household members continuing to undertake dog walking whilst exhibiting COVID-19 symptoms or self-isolating following development of symptoms. 

Responsibility for dog walking was often shared between household members—or before the pandemic, outsourced to a professional dog walker—to ensure that the dog’s needs are met. Professional dog walkers were significantly less likely to be called upon during lockdown, likely due to confusion regarding regulations, fear of contracting COVID-19, or people having more time to engage in dog walking themselves, potentially leading to health benefits for the owner but financial challenges for the professional dog walker. Nonetheless, both dogs and owners were more likely to have reduced dog walking duration during the first UK lockdown compared to their normal pre-lockdown routine if owners lived alone and/or considered themselves or someone in their household vulnerable to COVID-19. This may have been related to concerns by dog owners about risk of contracting COVID-19 during dog walking, similar to findings of other UK-based studies [17]. These findings were also consistent with previous UK research about the practical difficulties of dog walking for those who live alone [18], due to not having other household members who could walk the dog. Our findings suggest that restricting dog walking to once daily can have negative welfare impacts in particular for dogs with higher energy levels, or from multiple-dog households, as these dogs were walked significantly less during the lockdown compared to previously. In Montreal, Canada, the City imposed an 8pm curfew, allowing dog walking as the one exception outside of essential travel or emergencies, so long as walks remained within 1 km of home [36]. 

Respondents to the present study also adapted how and where they walked in accordance with government restrictions. Dogs spent more time on a leash and less time interacting with other dogs and people. Respondents travelled less to walk and dog walks occurred more often on streets, representing a shift from more recreational to functional walking locations [5]. Whilst this is likely to confer environmental benefits due to reduced pollution caused by driving [37], overcrowding in parks and green spaces was cited by some as a source of conflict between dog owners and other users, e.g., cyclists, people with children. To overcome this, some dogs were walked early in the morning or late at night, similar to findings from other UK-based research [17,24]. Future policy and programming should consider situations where dog walkers and other recreationists are expected to share common and different spaces, with focus on managing conflict and potential overcrowding. Most respondents believed that they had access to sufficient dog-supportive areas within walking distance from their house; however, opening areas normally not open to the public for dog walking (e.g., golf courses or school yards) during periods of high local demand for green spaces as well as educating park users about appropriate etiquette is advised.

In the present study, having a dog was considered by respondents to be a positive factor during the initial UK lockdown, providing companionship, purpose and motivation, a break from the bad, and a positive to focus on, all of which appears to be consistent with findings from other studies [17,19,38]. Respondents also reported on the value of physical contact with their dogs at a time where physical contact with others outside their household was prohibited, consistent with findings from another UK study where 90% of participants identified touch as a key way through which pets contributed to owner well-being [39]. Dog walking may also have been a key source of socially-distanced interaction with other people for those dog owners who lived alone. A small proportion of our respondents raised concerns regarding the potential risks of walking dogs. However, given research showing how beneficial dog walking is to individual physical health and low risk of contracting COVID-19 when outdoors, we believe that the benefits of walking the dog still outweigh the possible risk during this period of time.

Only 16.2% of respondents considered that owning and caring for a dog during lockdown had been challenging. This was a higher proportion than a recent UK study, where only 5% of respondents agreed with the statement “It would be easier for me not to have an animal at this time” [19]. Challenges that respondents stated included noticing behavioural changes in their dog or anticipating them, trying to follow public health guidelines while also meeting their pet’s needs, coping with crowding in parks, worrying about contracting COVID-19, and problems with accessing pet food and supplies, and related services (such as veterinary care, grooming, and dog day care). Similar challenges have been reported in other studies [40]. 

A separate set of challenges relates to providing care for a dog in case the owner is hospitalised. US-based research found that strength of pet attachment increased the odds of delaying treatment/testing for COVID-19; worrying about securing pet accommodation and organising care for their pet, were among the main reasons for delaying seeking help [40]. Similarly, when faced with floods, bushfires, or hurricane warnings, pet owners often delay evacuation and risk injury or even death by returning to their homes to rescue pets [41]. An emergency care plan for pets is therefore advisable, to improve the welfare of pets during pandemics and related emergencies and disasters and reduce health-risk to people such as delays in seeking help. 

Indeed, although a higher proportion of respondents living alone had an emergency care plan for their dogs, less than half of respondents had a care plan before the pandemic and just 7.7% developed one during the lockdown. Our findings suggest that further work is needed to both help owners prepare for inevitable future emergencies [42] and to provide support to owners at those times. Dog walkers could play an important part during emergencies by supporting the vulnerable, those living alone, and those with high-energy, active dogs, in supplementing dog walking. Local governments and not-for-profit organisations could further enable this support by providing dog walking discounts to those most likely to need additional help, or by setting up informal dog-walking networks. In addition, dog owners should be offered guidelines on how to prepare an emergency care plan for their dogs. Local governments and not-for-profit organisations could offer a website where such care plans can be stored and updated, in case the owner was not able to communicate this.

Our respondents reported on dog behavioural impacts that had occurred as a result of the lockdown restrictions (for example, increased reactivity whilst walking), or concerns in anticipation about them in future (for example when returning to work). A study of Spanish pet (dog and cat) owners also reported worsening of dog behaviour problems within only a few weeks of restrictions [23], a finding that was supported by UK data identifying small increases in growling/snapping/nipping, barking and whining, jumping up, and being clingy and following people around the house during lockdown restrictions [24]. In addition, studies have identified a spike in paediatric dog bites corresponding to introduction of lockdown measures [43,44]. Given that both adults and children would be spending more time with dogs at home during the lockdown restrictions [24] and that dog owners had noticed worrying behaviour changes in dogs, promoting educational advice to prevent dog bites is important, for example providing dogs a safe space within the house where they can be undisturbed. Findings from the present study suggest that whilst additional indoor activities with dogs helped with dog management, they were not a replacement for dog’s usual outdoor exercise. Nonetheless, in similar future situations where walking and outdoor exercise are restricted for dogs, owners could be advised on environmental enrichment that could be performed within the home (e.g., puzzle-feeders, scent and training games). Whilst there is evidence that some forms of indoor enrichment such as human–dog interactive games with a dog and training increased during lockdown compared to pre-lockdown levels, other forms such as provision of toys had not [24] and could be further promoted through dog welfare charities and national media.

The characteristics of dog’s and person’s dog walking prior to the pandemic were consistent with previous studies of factors associated with dog walking [9], lending validity to our data. Measuring the levels of physical activity through personal tracking devices is a more objective approach than reliance on respondents self-reports, and helped to validate self-reported changes in dog walking. The sub-population of owners who provided objective measures of steps taken (with and without their dogs) averaged 9996 steps per day. Further, owners reported on average 350 min of dog walking per week, which is likely to mean that most of the study respondents substantially exceeded the UK’s physical activity recommendations of 150 min of at least moderate physical activity per week [45]. This is in line with previous research findings that dog owners are more likely than non-owners to meet the physical activity guidelines [3]. 

Limitations of this study include the potential for a biased population of respondents compared to the wider UK dog-owning population: the majority of respondents were female, a high proportion were university-level educated, relatively few were classed as ‘vulnerable’ to COVID-19 and most had not developed any COVID-19 symptoms at the time of completing the questionnaire. In addition, as respondents participating in this study are likely to be selectively biased to those more committed to their animals, it is likely they walk more than some dog owners. The confidence intervals of the logistic regression models are broad, likely due to this study being underpowered, but results were consistent with those from other studies with larger sample sizes. Most data explored here were largely self-reported, which may introduce further uncertainty in our analysis. Future similar studies should investigate in more detail the impact of dog owners spending more time at home, being furloughed, or living in urban or rural areas. 

## 5. Conclusions

This study highlights the variability in changes in dog walking during the first national COVID-19 lockdown in the UK. We found that, overall, both dogs and people reduced the frequency of walks. However, the overall weekly duration of walks for dogs did not drastically change, likely because owners walked for longer and because in multi-person households, dogs had more people to support them. Some owners, in particular those living alone, self-identified as vulnerable, or owning high-energy dogs, had reduced dog walking options, which may have implications for both dog welfare and owner well-being. Having a dog provided benefits during lockdown; however, there were significant challenges to meeting the needs of pets and accessing care or supplies. Future provisions should be made to prevent and manage conflict between dog owners and other recreationists in parks and green spaces. Further, emergency preparedness, including care plans for pets, should be promoted, as future events or conditions that require others to take care of individual’s pets are likely. In order to optimise dog and owner welfare in similar lockdown situations, future policy should at least consider allowing dog owners who live alone and those with high-energy dogs to exercise their dogs more than once a day and allowing professional dog walkers to continue to work. Our findings support the English 3rd national lockdown decision to designate dog walking as unlimited, under exemptions for animal welfare and exercise [46]. However, whether this is widely realised by the public is not currently known.

## Figures and Tables

**Figure 1 ijerph-18-06315-f001:**
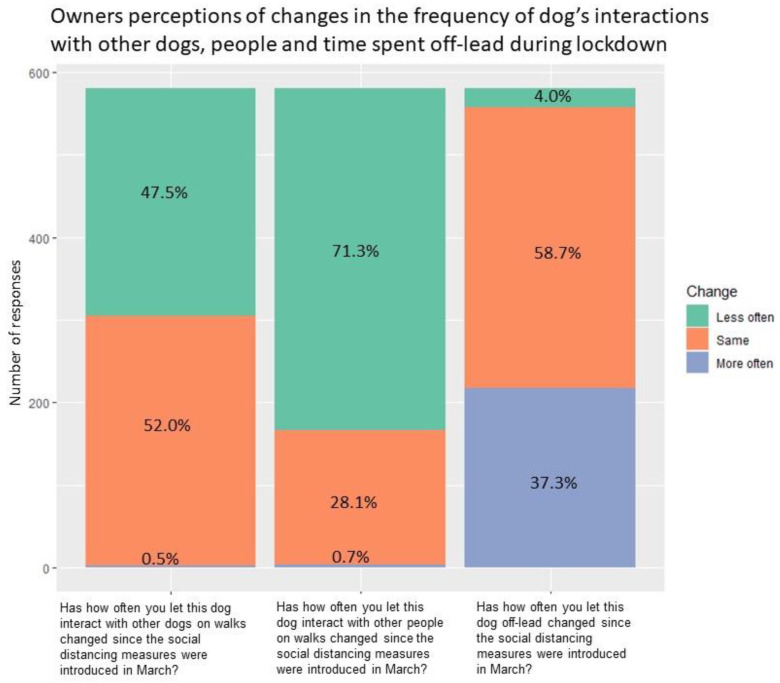
Owner’s perceptions of changes in the frequency of dog’s interactions with other dogs, people and time spent off-lead during lockdown.

**Figure 2 ijerph-18-06315-f002:**
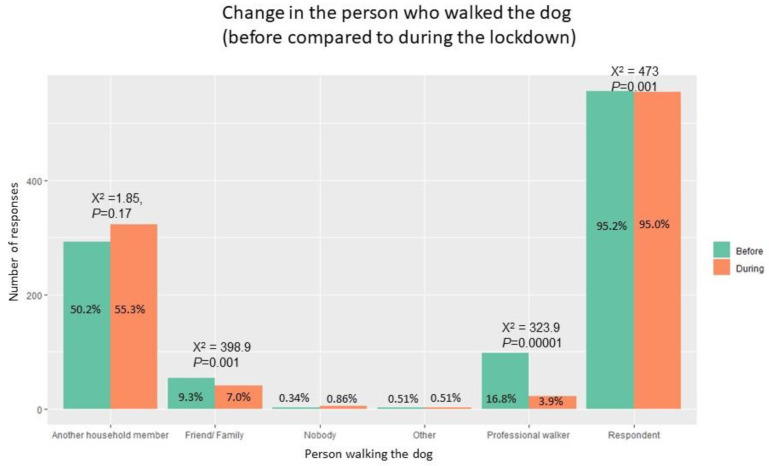
Person who walked the dog before and during the pandemic lockdown (McNemar’s X^2^ and *p*-value are provided where the sample size permitted a meaningful comparison).

**Figure 3 ijerph-18-06315-f003:**
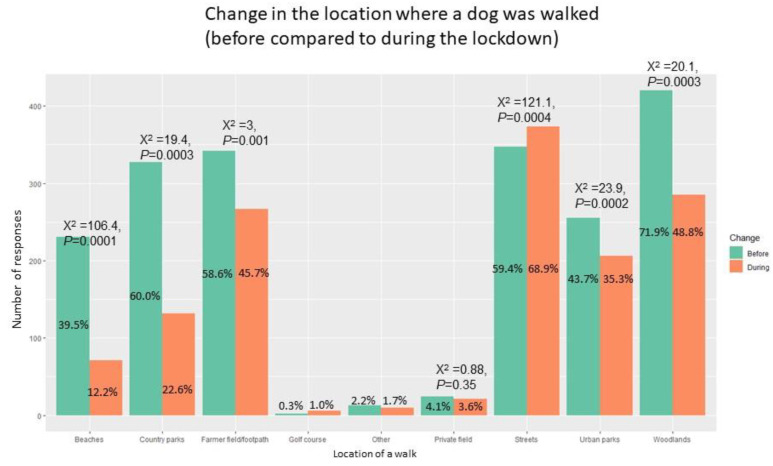
Walking location before and during the pandemic lockdown (McNemar’s X^2^ and *p*-value are provided where the sample size permitted a meaningful comparison).

**Figure 4 ijerph-18-06315-f004:**
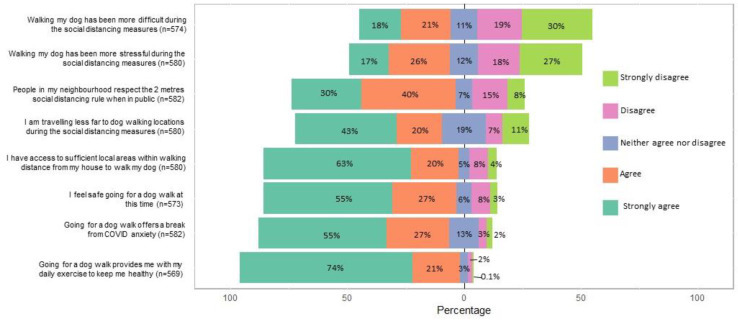
Summary of questions regarding experiences of and attitudes towards dog walking during the COVID-19 pandemic lockdown.

**Table 1 ijerph-18-06315-t001:** Weekly frequency of dog’s and person’s walking before and during the COVID-19 lockdown. Within-group comparison (change in the distribution of walking frequency between before and during the lockdown) is summarised with sample size (n), Mann–Whitney paired test-statistic (V) and significance levels (*p*).

Variable	Dog’s Walking			Person’s Walking		
	Before the Lockdown, (Median, IQR)	During the Lockdown, (Median, IQR)	Within-Group Change (n, V, *p*)	Before the Lockdown, (Median, IQR)	During the Lockdown, (Median, IQR)	Within-Group Change (n, V, *p*)
**Overall dog walking frequency**	10 (7)	7 (7)	558, V = 6665, *p* = 0.001	7 (9)	7 (1)	557, V = 139,706, *p* = 0.001
**Living arrangements:**						
Living alone	12 (7)	7 (7)	85, V = 3011, *p* = 0.04	10 (7)	7 (1)	86, V = 2980, *p* = 0.02
Living with others	10 (7)	7 (7)	463, V = 96,984, *p* = 0.009	7 (7)	7 (1)	462, V = 115,269, *p* = 0.03
**Household composition**						
Single-dog household	12 (7)	8 (7)	316, V = 45,196, *p* = 0.03	7 (9)	7 (3)	313, V = 46,513, *p* = 0.26
2 dogs	10 (7)	7 (7)	163, V = 11,598, *p* = 0.04	7 (9)	7 (1)	167, V = 11,637, *p* = 0.009
3+ dogs	7 (7)	7 (5.5)	79, V = 3527, *p* = 0.14	7 (7)	7 (1)	77, V = 3450, *p* = 0.07
**Dog size**						
Toy/Small	7 (7)	7 (7)	128, V = 7737, *p* = 0.42	7 (7)	7 (1)	126, V = 8562, *p* = 0.26
Medium	10 (7)	7 (7)	227, V = 284,611, *p* = 0.04	7 (7)	7 (2.75)	228, V = 28,497, *p* = 0.07
Large/Giant	12 (7)	7 (7)	199, V = 17,047, *p* = 0.013	7 (9)	7 (2)	198, V = 17,357, *p* = 0.04
**Dog’s age:**						
Less than 1 year	14 (7)	13 (7)	27, V = 320, *p* = 0.58	7 (4.5)	7 (2)	27, V = 346, *p* = 0.93
1–5 years	10 (7)	7 (7)	285, V = 37,047, *p* = 0.07	7 (9)	7 (1)	284, V = 37,438, *p* = 0.13
6–10 years	10 (7)	7 (7)	198, V = 16,765, *p* = 0.012	7 (8)	7 (2)	198, V = 16,544, *p* = 0.006
11+	10 (7)	7 (7)	46, V = 918, *p* = 0.25	7 (8)	7 (0)	46, V = 895, *p* = 0.42
**Dog’s energy levels:**						
High	12 (7)	7 (7)	207, V = 18,764, *p* = 0.023	7 (6)	7 (2)	207, V = 19,093, *p* = 0.05
Medium	9 (7)	7 (7)	310, V = 52,386, *p* = 0.04	7 (9)	7 (1)	308, V = 43,258, *p* = 0.069
Low	7 (7)	7 (7)	41, V = 759, *p* = 0.56	7 (8.5)	7 (1)	41, V = 754, *p* = 0.42
**Relationship with dog:**						
Weak	7 (3)	7 (3)	15, V = 109, *p* = 0.90	5 (3)	7 (3)	14, V = 103, *p* = 0.83
Medium	10 (7)	7 (7)	275, V = 33,906, *p* = 0.04	7 (7)	7 (1.5)	277, V = 35,049, *p* = 0.09
Strong	10 (7)	7 (7)	267, V = 31,615, *p* = 0.02	7 (8)	7 (3)	265, V = 30,807, *p* = 0.01

**Table 2 ijerph-18-06315-t002:** Between-group differences in frequency of dog’s and person’s walks per week both before and during the lockdown are summarised with Mann-Whitney test-statistics where 2 categories were compared (V), K-W test-statistic (X^2^) where 3 or more categories were compared and test significance levels (*p*). Where the between-group comparison was statistically significant (*p* < 0.05), Dunn *p* values with further Benjamini–Hochberg FDR adjustment for multiple comparisons are provided.

	Dog’s Walking		Person’s Walking	
Variable/Between-Group Difference (Test-Statistic, *p*)	Before the Lockdown	During the Lockdown	Before the Lockdown	During the Lockdown
Alone	12 (23)	7 (7)	10 (7)	7 (7)
Not alone	10 (24)	7 (7)	7 (7)	7 (1)
**Comparison**	V = 18,061, *p* = 0.09	V = 21,142, *p* = 0.4157	V = 21,142, *p* = 0.4157	V = 15,890, *p* = 0.0007
Single	12 (7)	8 (7)	7 (9)	7 (3)
2	10 (7)	7 (7)	7 (9)	7 (1)
3+	7 (7)	7 (5.5)	7 (7)	7 (1)
**Comparison**	K-W chi-squared = 4.8609, df = 2, *p* = 0.088	K-W chi-squared = 10.565, df = 2, *p* = 0.0051	K-W chi-squared = 2.0056, df = 2, *p* = 0.3669	K-W chi-squared = 2.5863, df = 2, *p* = 0.2744
		1 dog compared to 2: *p* = 0.029 1 dog compared to 3+: *p* = 0.18 2 dogs compared to 3+: *p* = 0.298		
Toy/small	7 (7)	7 (7)	7 (7)	7 (1)
Medium	10 (7)	7 (7)	7 (9)	7 (2.5)
Large	12 (7)	7 (7)	7 (9)	7 (2)
**Comparison**	K-W chi-squared = 1.7761, df = 2, *p* = 0.4114	K-W chi-squared = 0.36661, df = 2, *p* = 0.8325	K-W chi-squared = 1.6573, df = 2, *p* = 0.4366	K-W chi-squared = 1.5287, df = 2, *p* = 0.4656
Less than 1 year	14 (7)	13 (7)	7 (6.5)	7 (2)
1–5 years,	10 (7)	7 (7)	7 (9)	7 (1)
6–10 years,	10 (7)	7 (7)	7 (8)	7 (2)
11+	10 (7)	7 (7)	7 (8)	7 (0)
**Comparison**	K-W chi-squared = 3.1229, df = 3, *p* = 0.3731	K-W chi-squared = 5.221, df = 3, *p* = 0.1563	K-W chi-squared = 3.4335, df = 3, *p* = 0.3295	K-W chi-squared = 2.4675, df = 3, *p* = 0.4812
Energy low	7 (7)	7 (7)	7 (6)	7 (2)
Energy medium	9 (7)	7 (7)	7 (9)	7 (1)
Energy high	12 (7)	7 (7)	7 (8.5)	7 (1)
**Comparison**	K-W chi-squared = 3.7416, df = 2, *p* = 0.154	K-W chi-squared = 3.2972, df = 2, *p* = 0.1923	K-W chi-squared = 3.0876, df = 2, *p* = 0.2136	K-W chi-squared = 3.9702, df = 2, *p* = 0.1374
Relationship weak	7 (3)	7 (3)	5 (3)	7 (2)
Relationship medium	10 (7)	7 (7)	7 (7)	7 (1.5)
Relationship strong	10 (7)	7 (7)	7 (8)	7 (3)
**Comparison**	K-W chi-squared = 1.4272, df = 2, *p* = 0.4899	K-W chi-squared = 1.7668, df = 2, *p* = 0.4134	K-W chi-squared = 12.543, df = 2, *p* = 0.00189	K-W chi-squared = 9.0541, df = 2, *p* = 0.01081
			Weak compared to medium: *p* = 0.029 Weak compared to strong: *p* = 0.012 Medium compared to strong: *p* = 0.021	Weak compared to medium: *p* = 0.045 Weak compared to strong: *p* = 0.046 Medium compared to strong: *p* = 0.182

**Table 3 ijerph-18-06315-t003:** Between group differences for duration of walks both before and during the lockdown are summarised with Mann–Whitney test-statistic where 2 categories were compared (V), K-W test-statistic (X^2^) where 3 or more categories were compared and test significance levels (*p*). Where the between-group comparison was statistically significant (*p* < 0.05), Dunn *p* values with further Benjamini–Hochberg FDR adjustment for multiple comparisons are provided.

Variable/Between-Group Difference (Test-Statistic, *p*)	Dog’s Walking before the Lockdown	Dog’s Walking during the Lockdowns	Person’s Walking before the Lockdown	Person’s Walking during the Lockdown
Alone	500 (500)	420 (330)	440 (420)	420 (335)
Not alone	420 (350)	420 (300)	300 (300)	315 (210)
**Comparison**	V = 16,524, *p* = 0.06	V = 18,315, *p* = 0.72	V = 13,254, *p* = 0.000048	V = 14,129, *p* = 0.0002515
Single dog	420 (398)	420 (330)	300 (300)	325 (214)
2 dogs	420 (320)	420 (310)	400 (360)	365 (285)
3+	420 (388)	420 (335)	420 (405)	420 (350)
**Comparison**	K-W chi-squared = 1.0261, df = 2, *p* = 0.5987	K-W chi-squared = 2.2545, df = 2, *p* = 0.3239	K-W chi-squared = 9.448, df = 2, *p* = 0.00888	K-W chi-squared = 2.5863, df = 2, *p* = 0.2744
Toy/small	420 (290)	420 (285)	300 (315)	345 (220)
Medium	420 (350)	420 (350)	360 (368)	400 (210)
Large	468 (400)	420 (330)	350 (370)	355 (282)
**Comparison**	K-W chi-squared = 4.8343, df = 2, *p* = 0.08917	K-W chi-squared = 1.2931, df = 2, *p* = 0.5238	K-W chi-squared = 1.4663, df = 2, *p* = 0.4804	K-W chi-squared = 0.56278, df = 2, *p* = 0.7547
Less than 1 year	420 (345)	480 (280)	350 (315)	345 (248)
1–5 years	420 (400)	420 (330)	315 (360)	360 (270)
6–10 years	420 (330)	420 (322)	360 (370)	360 (280)
11+	400 (220)	420 (250)	300 (225)	280 (220)
**Comparison**	K-W chi-squared = 4.2801, df = 3, *p* = 0.2328	K-W chi-squared = 4.1946, df = 3, *p* = 0.2412	K-W chi-squared = 2.0934, df = 3, *p* = 0.5532	K-W chi-squared = 2.399, df = 3, *p* = 0.4938
Energy low	300 (292)	338 (195)	225 (260)	205 (195)
Energy Medium	420 (320)	420 (320)	340 (345)	360 (270)
Energy High	540 (490)	420 (350)	360 (390)	402 (255)
**Comparison**	K-W chi-squared = 25.415, df = 2, *p* = 3.028 × 10^–6^	K-W chi-squared = 18.846, df = 2, *p* = 8.083 × 10^–5^	K-W chi-squared = 12.601, df = 2, *p* = 0.001835	K-W chi-squared = 22.548, df = 2, *p* = 0.0000127
	Low compared to high: *p* = 0.00018 Low compared to medium: *p* = 0.04187 Medium compared to high: *p* = 0.0000000005	Low compared to high: *p* = 0.00025 Low compared to medium: *p* = 0.00610 Medium compared to high: *p* = 0.00610	Low compared to high: *p* = 0.0023 Low compared to medium: *p* = 0.0087 Medium compared to high: *p* = 0.0838	Low compared to high: *p* = 0.000000006 Low compared to medium: *p* = 0.000027 Medium compared to high: *p* = 0.24
Relationship weak	420 (210)	420 (200)	250 (65)	300 (108)
Relationship medium	420 (350)	420 (305)	300 (285)	310 (210)
Relationship strong	420 (400)	420 (330)	420 (420)	420 (290)
**Comparison**	K-W chi-squared = 0.9277, df = 2, *p* = 0.6289	K-W chi-squared = 0.32669, df = 2, *p* = 0.8493	K-W chi-squared = 10.397, df = 2, *p* = 0.005524	K-W chi-squared = 5.4787, df = 2, *p* = 0.06461
			Weak compared to strong: *p* = 0.047 Weak compared to medium: *p* = 0.017 Medium compared to strong: *p* = 0.231	

**Table 4 ijerph-18-06315-t004:** Multivariable logistic regression model for dog’s reduction in total weekly dog walking duration during the COVID-19 pandemic lockdown. Model based on 514 observations, X^2^ = 6.72, *p* = 0.03, AIC = 642.98.

Variables	Categories	Odds Ratio (95% Confidence Interval)	Z Value	*p*
Owner’s living arrangements	Owner does not live alone	1	-	-
	Owner lives alone	1.61 (0.99–2.63)	1.90	0.06
Owner/household vulnerability to COVID-19	Owner or household members not described as vulnerable	1	-	-
	Owner or household member considered vulnerable	1.55 (1.00–2.40)	2.01	0.04

**Table 5 ijerph-18-06315-t005:** Multivariable logistic regression model for reduction in person’s total weekly dog walking duration during the COVID-19 lockdown. Model based on 489 observations, X^2^ = 7.67, *p* = 0.02, AIC = 572.63.

Variables	Categories	Odds Ratio (95% Confidence Interval)	Z Value	*p*
Owner’s living arrangements	Owner does not live alone	1	-	
	Owner lives alone	1.83 (1.09–3.07)	2.30	0.02
Owner/household vulnerability to COVID-19	Owner or household members not described as vulnerable	1	-	
	Owner or household member considered vulnerable	1.55 (0.97–2.46)	1.87	0.06

**Table 6 ijerph-18-06315-t006:** Qualitative analysis of respondents answers regarding the benefits of owning and caring for a dog during the COVID-19 lockdown.

Theme	Sub-Themes and Supporting Quotes
Companionship	*Reduces feelings of isolation and loneliness* • I live alone so my dog has been great company for me. • Felt less isolated as saw other dog walkers I knew when out and about. *Provides comfort and physical affection* • Can’t hug or kiss or touch other people outside the household so dog is an extra cuddle. • The physical contact and company have been comforting.
Purpose and motivation	*Reason to get up and get dressed in the morning* • Gives purpose and structure to the day. • [My dog] is the sole reason I get out of bed in the morning. *Reason to get outside and keep active* • I probably wouldn’t have gone out at all, other than to the shops, if it weren’t for the dog. • Obviously I could have walked without a dog, but I don’t think I would have every day.
Break from the bad	*Distraction from the news* • Takes my mind off COVID-19 issues. • I have somewhere to focus my energy that’s not on news and bickering and the terrifying death toll. *Break from each other, when everyone at home* • Allowed me to get out the house and escape the family. • I live with a partner who is shielding and suffers from anxiety. My dog allows me to focus on something other than my partner and his worries during this time. *Break from the computer, when working from home* • Distraction from working from home so I am not sat in front of the computer constantly. • When working from home, daily walks have been great for taking a break and getting fresh air. *Break from others on walks, giving some dogs the space they need* • Social distancing has been advantageous [for us] on walks because my dog can be reactive. • The 2 m distance from people is brilliant for me and my dog. We are much more relaxed knowing people aren’t going to approach my dog.
A positive to focus on	*Helped keep things in perspective* • Caring for [the dogs] has helped keep things in perspective. • [Dogs] have made me see the “bigger picture”. *Reminder to live in the moment* • Dog walking is the best thing in the world to unwind and be in the moment. • I have explored new tracks and really appreciated fully the abundance of wildflowers and birds in the area that I live. • He has so much joy in a walk it is impossible not to love watching him. *Constant amidst change* • Everything else in our lives has changed; however, the need to care for our dog remains constant. • Bit of normality that has continued from pre-lockdown times. *Something other than virus to talk about* • A nice topic to talk to family and friends about. • Cuteness of our dogs is often a conversation starter and distracts from the lockdown melancholy many people are experiencing. *Feelings of joy and laughter* • Brought fun and laughter to the household [during] otherwise depressing time.

**Table 7 ijerph-18-06315-t007:** Challenges of owning and caring for a dog during the period of isolation/distancing.

Theme	Sub-Themes and Supporting Quotes	
Dog behavioural impacts	Actual	*Becoming reactive on walks* • [Dog] picks up on the atmosphere when we meet people who move away from us and is scared. She has never been like this before. • Dogs are starting to bark at approaching people due to me avoiding them. *Becoming restless and needy at home* • Dog now thinks that anytime you put on shoes or a jumper you are taking him for a walk. • Dog sits and stares at me while I’m [working from home] and begs for me to go downstairs by pawing at my leg. *Barking at home at occurrences outdoors* • More neighbours doing DIY [do it yourself] etc. making noise causing upset barking dogs. • More deliveries coming to the house and knocking on the door. For example, groceries. Dogs go bonkers. • Barking behaviour in garden stressful, especially when surrounding neighbours are now always at home too.
	Anticipated	*Potential for separation anxiety, once households return to work and school* • I am concerned about them coping when I am back in the office […] I am ensuring dogs are getting their own space and used to people not being around all the time when I can. • Having a [puppy] at start of lockdown and a wife having to shield meant we haven’t been able to do any training to help him cope with future separation from us for a couple of hours. This leaves us unsure how he will cope and adjust having always had someone around. *Potential for dogs to become wary of others, due to reduced opportunities for socialisation* • I worry that people’s wariness of interacting and our lack of usual sociable group walks will negatively affect how my dog interacts with other dogs after lockdown. • I got a planned puppy just before lockdown and it’s been difficult in socialising and taken them to different places/events. *Potential for dog bites, with people always at home* • Lockdown with the toddler has put pressure on the dog. It’s a constant battle to get him the space he needs to sleep.
Balancing public health guidance with meeting the needs of their dog	Meeting physical needs of dog with 1-walk-a-day rule	*Needing more, especially higher-energy dogs* • Challenging when only allowed one hour of daily exercise due to having a very active dog. I walk dog in the mornings, husband does afternoons. • I live in an apartment with no outside space so could not respect the 1-walk-a-day. Normally the dog would walk three times a day and go agility training three times a week. • My dog is used to considerably longer walks than what she has been getting and maintaining her fitness hasn’t been easy. She isn’t overly found of games in the house and garden-we have tried things like scent work at home and she just didn’t want to know.
	Meeting mental needs of dog when trying to stay as close to home as possible	*Becoming bored with same routes/scenery every day* • I previously drove or took public transport to bigger spaces to exercise [my dog] but have been mainly stuck with a relatively small local park. • We do less variety of walks, so they are becoming repetitive. • My dog got very bored of the same walk from home and on occasions didn’t want to go out.
	Meeting social needs of dog when trying to maintain 2 m distance outdoors	*Social dogs and owners having difficulties adjusting* • Social aspect of dog walking stopped completely-one of the main pleasures. • I have found it really emotionally difficult to not pat other dogs that have come up to me for attention when walking my dog. It’s really hard as I feel like I’m rejecting them and they don’t understand why I’m not patting them. It’s the same when I see my dog not getting attention from other people. I know it’s the right thing to not have contact, but it made walks more difficult and left me feeling a bit guilty. • My dog has been separated from his dog friends. • My dog misses seeing people we know.
Conflict in outdoor recreation	Crowding	*Newcomers and nicer weather* • I felt less comfortable going to my usual walking routes due to too many people going to the same place. • Walking at busy times and on sunny days was very stressful due to other people not respecting social distancing. *Judgment towards dog owners (lack of control, off-leash dogs who are poorly socialised, not picking up after pet)* • Huge increase in number of dogs around. Maybe this is because people’s work pattern has changed so they are walking them at different times, but it does worry me that maybe there are lots of dogs not usually walked? • Far more people out walking their dogs, who frankly don’t have a clue about walking their dog. Dogs off-leash, no recall. Walking with headphones on or on their mobile phones with no awareness of their dog. • Some people haven’t bothered to keep their dogs under control and I’ve seen more puppies than ever. • Poorly socialized. Many lost dogs in the area....not used to being let off-leash. • Dog mess is not cleaned up and has increased, with regular dog walkers in the area bearing the brunt of the backlash of this. *Judgement towards other recreationists (joggers, cyclists, families with children)* • Parks are much busier with non-dog-walkers, litter and food waste much worse. • The most stressful part of dog walks are all the extra families with children who have no dog/outdoor sense, and a general lack of awareness and don’t/won’t follow social distance rules. They seem to think they have priority over everyone else just because they have a child, even though they don’t normally use any of the outdoor spaces like the rest of us (dog walkers/riders/runners/cyclists) do.
	Coping	*Walking during off-peak hours and other modifications* • The change I made was to walk much earlier during the day with my dogs, to avoid other people. • I gave up trying to walk in my usual green spaces as it was just too scary and busy. • Dangerous when having to walk on the road to avoid people. • He is reactive and there’s too many people out. He can only be walked at midnight due to this.
Contracting COVID-19		*Potential for virus to be left behind on infrastructure in parks and public space* • The only thing that stresses me is the poo bins, loads of people use them and you have to use your hands to open them. I can’t help but always think how many people don’t sanitise or was their hands before/after using them. • Have not noticed anyone taking precautions (i.e., carrying hand sanitiser, washing hands) after touching gates. *Virus transmission through dog’s fur or saliva* • Some people also seem to think they can let their dogs socialise freely with mine if their leashes are long enough, and I don’t want to take the small risk that their dog may have COVID on their fur or in their saliva. • When someone’s dog attacked mine, she came right up to me when I was checking my dog over and grabbed my dog. I was surprised by how concerned I was about her handling my dog and being in my space. I ended up showering my dog when I got home because she had her hands all over him. I am not like that!
Accessing pet food, supplies, and services		*Difficulty finding dog food and supplies* • Ensuring I can get food, medication or toys without car and avoiding taking public transport. • Dog food stock shortages [when] people panic buying. • Harder to get special diet dog food from shop. *Veterinary visits, esp. for older dogs or those with pre-existing conditions* • Main stress has been access to vet/restriction in end-of-life care for another dog of mine that passed away during lockdown. • Concerned that if there was a dog fight or other emergency from our walks we wouldn’t easily be able to go to a vets. • He needs physio & hydro […] we haven’t had access to our vets as they are only seeing emergency patients so we struggle on with symptoms and less support. We have also had to stop laser treatments for pain. • Harder to get to the vet. One dog ate something that got stuck. Had to phone when outside, then wait as people already in practice. Then had to leave dog in consult room whilst vet came in from other side and took dog out the back. Appreciate the measures in place, just so foreign especially when dog in distress. *Requiring grooming or day care, yet businesses closed* • My dog requires regular stripping to keep her coat healthy. I am not able to do it as my hands are arthritic. My groomer had to close and so dog is hot, hairy, and uncomfortable. Not fair on her and she doesn’t understand. • Having to worry more at times about having someone look after my dog when I am at work as dog day care has been closed. • The dog walker helped him socialise within a carefully selected pack and under supervision (he will nip or fight if he feels threatened due to past trauma). Since lockdown they haven’t been able to go out with the walker and I worry about whether this has set him back in his progress.

## Data Availability

As free-text boxes may contain data enabling participant identification, data is not freely available.

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
