# Peer review of "Dog Walking before and during the COVID-19 Pandemic Lockdown: Experiences of UK Dog Owners"

_ijerph, 2021, doi:10.3390/ijerph18126315_

Round 1

Reviewer 1 Report

The aim o the study is to investigate the impact of first COVID-19 UK lockdown on dog walking and ownership. An online survey was developed to collect the data end it was distributed via social media.

The issue sounds very interesting and it is a current topic. The lockdown has impacted on different levels in human and pet life. Explore this aspect is important in order to implement adequate strategy to address another pandemic event or similar.

Introduction

LL 53-55 this can be used in the discussion. Add reference

Materials and Methods

Generally, the materials and methods should be revised. In particular the survey design because it is hard to understand the results. I guess the questionnaire was divided in sections and each section contended questions related to specific information (e.g. owner demographics; dog demographics, living environment etc.) I suggest to describe the questionnaire considering the parts in which is divided adding a brief resume of the type of questions (possible add this in supplementary materials).

LL 123-124 specify that these data were collected only in a subsample of owners. And also specify the kind of device (app mobile or dedicated device)

LL 124-125 specify what it means “perception around the dog”

LL 131 Which close-ended questions? Put some examples

LL 131-132 Did open ended include dog behaviour (perceive challenges)?

LL 137-138 it is not totally clear. This seems more a result. Please explain better this point.

LL 138-139 It is not clear. I suggest to rephrase this sentence. Did the authors find some contradictory answers in people who defined themselves as living alone? In which other way in the questionnaire people living alone could be identified?

LL 140-141 please verify this sentence. It is not clear how these answers were merged

LL 172-174 please specify this aspect also in the survey design part (questionnaire)

LL 179 explain the acronym (Table maintenance generator?) it could be interesting to explain better this approach

Results

In my opinion it is very hard reading and consequently understand the results. There are many results and some tables are very big. I suggest to focalise the manuscript on the main and significant results and reduce the tables putting them in supplementary materials (e.g. table 11, 12).

In this case the table are not totally supporting text to understand the results but make reading quite difficult. If you write the result in the text it is not necessary repeat it in a table and vice versa.

LL 206 please explain better through some examples what you mean for emergency care plan for dogs. Also, this issue is not in materials and methods in the questionnaire description.

LL 215 it is not clear how the authors discriminated the level of energy. How is the question about this? Did you give some definition to owners? Or is it only the perception of owner?

LL 234 The same for the definition of relationship strength. How is the question? This categorization is based on scientific literature?

Table 2 vs table 3 are used the terms before the lockdown, during lockdown in the same manner of before the pandemic, during pandemic? I suggest to homologate the terminology. Please check this in all table (e.g. table 5, table 4 etc.)

LL 253-254 this should be move in the discussion

LL 280-289 it could be useful adding graphs to summarise these kind of data

LL 346-354 are these results from TMG? (see also LL 362-369 here the authors put some percentages)

LL 369-370 Please specify the analysis used to obtain this result

LL 375 378 Sorry but I do not understand how did the authors calculate these percentages? The total respondents considered for the analysis is 584. I suppose that for this point the authors have considered a sub-sample of respondents. Please clarify this aspect.

LL 399-401 The authors should describe better in the materials and methods how to the theme were identified. The same for LL 413-417

Table 1 I suggest to move in the supplementary materials (the same for table 12)

In my opinion, the data on behaviour of dogs deserves a better analysis. If it is possible it could be interesting exploring the correlation between walking duration and behavioural impact on dog, level of energy etc. From this point of view, it is possible to better discuss the result (see LL 445-448)

In table 12 please explain the acronym of DIY

Discussion

LL 445-448 as said previous, I think that it is not totally correct to support this affirmation by a descriptive analysis. Please clarify this point.

LL 489-491 this paragraph seems disconnected to the previous one.

LL 494 not only to improve animal welfare but also for a correct management of pets when the owner is not able to take care and to address their needs.

LL 523-525 Sorry but I cannot find this point in the results.

In general, I suggest to revise the discussion according to the results description. The first results presented were on the walking duration and frequency and also the discussion should be done following the same structure, in this way it is easier reading and understanding. The focal point of this study is changing in walking during pandemic but the discussion about this item seems poor and not central.

Author Response

Thank you for providing us with constructive comments and suggestions. Below we summarise how these were addressed.

Introduction

LL 53-55 this can be used in the discussion. Add reference- The statement helps to justify why this study was conducted, which is why we would like to keep it in the introduction. References to the current guidelines was added.

Materials and Methods

Generally, the materials and methods should be revised. In particular, the survey design because it is hard to understand the results. I guess the questionnaire was divided into sections and each section contended questions related to specific information (e.g. owner demographics; dog demographics, living environment etc.) I suggest to describe the questionnaire considering the parts in which is divided adding a brief resume of the type of questions (possible add this in supplementary materials).

We have now expanded the details of questions included in the survey (lines 116-170). This section reads:

The questionnaire consisted of 64 open- and close-ended questions, divided into 8 sections? : 

  • “About your dog ”: number of dogs in the household; dog size; age; sex; dog’s general energy levels as perceived by the respondent (i.e. no further definitions were provided) and owner-perceived relationship with their dog (based on the Inclusion of the Other in the Self question adapted for pets [28, 29]. Respondents were presented with seven images of circles with a progressing degree of an overlap between a circle representing a dog and an owner: starting with completely separate circles, representing a weak relationship, and ending with two nearly completely overlapping circles, representing a strong relationship). For questions relating to only one dog, respondents were asked to answer based on the dog they felt emotionally closest to. This strategy was chosen because dog walking is known to be associated with the strength of the relationship between the owner and the dog [9], and therefore this was the dog most likely to be walked pre-lockdown and thus affected by it.
  • “Walking this dog”: frequency of dog’s interactions with people and other dogs on walks before the lockdown; dog’s perceived recall reliability (response categories: dog never comes back when called, rarely, sometimes, often); weekly frequency and duration (in minutes) of dog walking undertaken by dog(s) during the lockdown and before (based upon the Dogs And Physical Activity tool [30]); perceived changes in total number of walks a dog gets from anyone since the lockdown; and perceived changes in how often a dog is off-lead, allowed to interact with other dogs and people since the social distancing measures were in place.
  • “Who walks dogs”: who walked the dog during lockdown and before; whether the person who walks the dog changed since the lockdown; and if so, how (open-ended question).
  • “Personal dog walking”: weekly frequency and duration (in minutes) of dog walking undertaken by the respondent during lockdown and before [30]; perceived changes in respondent’s number of dog walks since the lockdown and daily number of steps taken pre and during the first national lockdown (from respondent’s own recording devices, such as mobile phones or smart watches).
  • ”Other dog walking”: location of dog walking both before and since the lockdown; perceived changes to walking location; description of these changes (open-ended question), whether COVID-19 changes brought the respondents into greater contact with livestock when walking; and attitude-related questions regarding experience of dog walking during the lockdown (e.g. Going for a dog walk offers a break from COVID anxiety. Responses to these questions were presented on a 5-point Likert scale anchored with ”strongly agree” and “strongly disagree”.
  • “Perceptions of dog ownership”: whether caring for a dog during the period of social isolation been challenging and helpful; two open-ended questions enquiring about ways in which caring for a dog was challenging and helpful during the lockdown; presence of an emergency care plan (defined in the survey as “verbal or written agreement about who would care for your dog if you were ill or other plans for an emergency”) and whether the respondent made one since the coronavirus outbreak.
  • “COVID questions”: whether the respondent experienced suspected COVID-19 disease and if so, whether they and household members walked their dog during the period when they had symptoms and when symptoms were not present any more but they were still within the designated isolation period; whether the respondent or household member was vulnerable and told to isolate for 12 weeks regardless of the symptoms and whether the vulnerable person continued to walk the dog whilst isolating.
  • “About you”: respondents age, gender, qualifications, and an open-ended questions about anything else regarding experience of dog walking during the coronavirus outbreak.

LL 123-124 specify that these data were collected only in a subsample of owners. And also specify the kind of device (app mobile or dedicated device)- This data was collected from all respondents, but only some respondents answered this question. This is clarified in text in line 391 (“Among the owners who provided a daily step count based on personal activity trackers (n= 106, 18.2% of all respondents)…) The type of devices respondents may have used is now specified in line 146-147

LL 124-125 specify what it means “perception around the dog”-  Details of questions included in this section are provided in lines 156-162.

LL 131 Which close-ended questions? Put some examples- examples provide in line 172-176: Some open-ended questions included answer choice ‘Other- please specify’. These answers were re-coded where possible to match the pre-specified answers (e.g. in local woods was re-coded into pre-existing category “Woodlands”), or a new category was created (e.g. in response to the question about where dogs were walked during the lockdown, a category “private field” was created).  

LL 131-132 Did open ended include dog behaviour (perceive challenges)?- Yes if that is what the respondent thought to write. Perceived challenges were coded qualitatively, as explained in the section on qualitative coding (lines 234-239). Examples of re-coded answers are provided in lines 172-176.  

LL 137-138 it is not totally clear. This seems more a result. Please explain better this point.- This sentence explains how data was handled and for this reason we would like for it to remain in the methods section. It has now been re-written to improve clarity (Lines:180-183 “As a part of checking the data for imputation errors, in line with previously published cut-off points, weekly walking duration was capped at 2520 minutes maximum (approximately 6 hours a day) [31]. Values greater than this were considered as entered in error and were converted to 2520 ahead of the analysis”.  

LL 138-139 It is not clear. I suggest to rephrase this sentence. Did the authors find some contradictory answers in people who defined themselves as living alone? In which other way in the questionnaire people living alone could be identified? This has been clarified in lines 183- 187 (“People living alone were identified by filtering those who answered “I live alone” to the question about other household members experiencing COVID-19 symptoms and the question about other household members being classed as vulnerable. Contradictory answers (e.g. answering “I live alone” to the first question and “Yes [other household members are vulnerable]” to the second one) were not used in the analysis.) and in the results in lines 255-257 (“Six respondents provided contradictory answers to the question to which “I live alone” was one of the potential responses. These data were removed and a resulting 15.5% (n=89) respondents were identified as living alone”)

LL 140-141 please verify this sentence. It is not clear how these answers were merged. To improve readability, these results are now presented as a figure (Figure 4) and in full, therefore the explanation about merging was removed.

LL 172-174 please specify this aspect also in the survey design part (questionnaire)- this is now explained in lines 166-172.

LL 179 explain the acronym (Table maintenance generator?) it could be interesting to explain better this approach- these are the initials of the co-author who carried out the qualitative analysis. We clarified this sentence by adding: one of the co-authors (TMG) (lines 231-233): “Thematic analysis was carried out. Answers to open-ended questions were coded line-by-line by one of the co-authors (CW or TMG), by assigning codes that captured the main sentiments conveyed within and across responses [35]”.

Also added: Initial codes were revised throughout the coding process to ensure codes accurately REFLECT the text. Codes were then compared and grouped and patterns emerging from the initial coding were discussed between the co-authors  and after further revision the main themes discussed and identified [35].

Results

In my opinion it is very hard reading and consequently understand the results. There are many results and some tables are very big. I suggest to focalise the manuscript on the main and significant results and reduce the tables putting them in supplementary materials (e.g. table 11, 12). In this case the table are not totally supporting text to understand the results but make reading quite difficult. If you write the result in the text it is not necessary repeat it in a table and vice versa.

As this is largely an exploratory analysis, we do not have a single result to focus on. Tables 11 and 12 (now 8 and 9) are the only evidence we provide to illustrate the qualitative analysis. As the standards for reporting qualitative data (e.g. COREQ) specifies provision of quotations to illustrate the main themes, we would prefer not to remove these tables. We have however implemented other changes in the way the results are presented:

Table 4 (Weekly duration of dog’s and person’s walking before and during the COVID-19 first national lockdown) was moved into Supplementary material as the findings discussed there are not significant.

Two tables summarizing multivariable regression (data which provides context for dog walking before the pandemic) were moved to the supplementary material

Tables 9 (Person who walked the dog before (pre- first lockdown) and during the pandemic (during first lockdown)  and 10 (Walking location before (pre- first lockdown) and during the pandemic (during first lockdown) were converted into Figures 2 and 3.

We understand that where the results based on closed-ended and open-ended questions were discussed together, it may have appeared as repetitive. Summary of closed-ended questions regarding experiences and attitudes to dog walking is now presented as a figure (Figure 4) and separate from findings of qualitative analysis. Results derived from the qualitative analysis is preceded with words that suggest that the results were identified in this way (e.g. “Qualitative analysis of open text responses indicated that owners stopped using professional dog walking/care services or informal friends and family, because they were not allowed or because they did not need it any more as somebody else was able to walk the dog (working/studying from home, or being on furlough”, line 397). We hope these amendments improve the readability of the text.

LL 206 please explain better through some examples what you mean for emergency care plan for dogs. Also, this issue is not in materials and methods in the questionnaire description. This question is now explained in the methods section (line 162-165). As this was a close-ended question (with yes/ no offered as an answer), we cannot add examples of emergency care plans developed owners in this study.

LL 215 it is not clear how the authors discriminated the level of energy. How is the question about this? Did you give some definition to owners? Or is it only the perception of owner? –The energy levels were based on owner’s perceptions. This is now clarified in the questionnaire description (line 123-124).

LL 234 The same for the definition of relationship strength. How is the question? This categorization is based on scientific literature? This was explained with reference to existing studies in the original draft in lines 119-121. A further clarification has now been added (lines: 126-130). Note on how this data was recoded ahead of the analysis is added in lines 214-215.  

Table 2 vs table 3 are used the terms before the lockdown, during lockdown in the same manner of before the pandemic, during pandemic? I suggest to homologate the terminology. Please check this in all table (e.g. table 5, table 4 etc.)- Thank you for drawing our attention to this. We have now revised this and to be consistent the term before/after lockdown was selected.

LL 253-254 this should be move in the discussion – this has been removed from the results and as it was already present in the discussion it was not added there.

LL 280-289 it could be useful adding graphs to summarise these kind of data- this was converted into Figure 1.

LL 346-354 are these results from TMG? (see also LL 362-369 here the authors put some percentages)- The percentages refer to closed-ended questions whilst the remaining results come from the qualitative analysis. We have now presented the results of close-ended questions in Figure 2 and signposted the reader to these findings in the text and clarified that the remaining results are derived from qualitative analysis (Line 397) :” Nearly a third of respondents (28.8%, n=168) stated that a different person was walking their dog compared to pre- lockdown. Compared to before, during lockdown, significantly fewer people used dog walkers (P<0.001) or other friends/family (P<0.001) and more walked the dog themselves (P<0.001) (Figure 2). Qualitative analysis of open text responses indicated that owners stopped using professional dog walking/care services or informal friends and family, because they were not allowed or because they did not need it any more as somebody else was able to walk the dog (working/studying from home, or being on furlough…”

. LL 369-370 Please specify the analysis used to obtain this result- This is added in lines 209-211: “Comparison of where a dog was walked and presence/ absence of an emergency care plan between those who did and did not live alone was carried out with a Chi-square test.”.

LL 375 378 Sorry but I do not understand how did the authors calculate these percentages? The total respondents considered for the analysis is 584. I suppose that for this point the authors have considered a sub-sample of respondents. Please clarify this aspect. We added the denominator described earlier in the text (line 202) in this paragraph for clarity (lines 388-390): “Among the owners who provided a daily step count based on personal activity trackers (n= 106, 18.2% of all respondents), the mean number of steps taken daily…”

LL 399-401 The authors should describe better in the materials and methods how to the theme were identified. The same for LL 413-417. Qualitative analysis, including coding process is described in (lines 231-237) and more details were added: “Thematic analysis was carried out. Answers to open-ended questions were coded line-by-line by one of the co-authors (TMG or CW), by assigning codes that captured the main sentiments conveyed within and across responses [35]. Initial codes revised throughout the coding process to ensure codes accurately reflex the text. Codes were then compared and grouped and patterns emerging from the initial coding were discussed between the co-authors. The main themes were identified by revising the grouping of the initial codes [35].”

Table 1 I suggest to move in the supplementary materials (the same for table 12) Tables 11 and 12 (now 8 and 9) are the only evidence we provide to illustrate the qualitative analysis. As the standards for reporting qualitative data (e.g. COREQ) specifies provision of quotations to illustrate the main themes, we would prefer not to remove these tables.

In my opinion, the data on behaviour of dogs deserves a better analysis. If it is possible it could be interesting exploring the correlation between walking duration and behavioural impact on dog, level of energy etc. From this point of view, it is possible to better discuss the result (see LL 445-448)- We did not collect the data on dog behaviour (e.g. energy levels) before the pandemic; owners were asked to assess their dog’s normal energy levels in general. Therefore, we cannot analyse how the lockdown restrictions impacted on this facet of dog behaviour. We have however analysed qualitative text on this subject (presented in Tables 8 and 9). The only data for which we collected information for both before and since the pandemic was frequency and duration of walks. Dog (and owner’s) demographic variables, including dog’s energy levels were included in the models predicting reduction (compared to maintenance) of walking duration during lockdown.

In table 12 please explain the acronym of DIY- explanation (Do it Yourself) is now added (Table 9)

Discussion

LL 445-448 as said previous, I think that it is not totally correct to support this affirmation by descriptive analysis. Please clarify this point. This point emerged from Kruskal Wallis and Mann-Whitney test, not descriptive statistics. As explained above, as we did not collect data on this aspect of dog behaviour before and since the first national lockdown in the UK, we cannot carry out analysis that could clarify quantitatively how dog behaviour has changed.

LL 489-491 this paragraph seems disconnected to the previous one. This sentence has been split into a separate paragraph and clarifying text was added (lines 539-551):

 “…Similar challenges have been reported in other studies [40].

              A separate set of challenges relates to providing care for a dog in case the owner is hospitalised. US-based research found that strength of pet attachment  increased the odds of delaying treatment/ testing for COVID-19; worrying about securing pet accommodation and organising care for their pet, were among the main reasons for delaying seeking help [40]. More broadly, research shows that owner-pet relationship affects owner’s behaviour in the context of facing a natural or man-made disasters, leading to a compromise in public health. For example,  when faced with floods, bushfires or hurricane warnings, pet owners often delay evacuation and risk injury or even death by returning to their homes to rescue pets [41].  An emergency care plan for pets is advisable to improve welfare of pets by ensuring they are correctly managed and reduce health-risk to people by reducing delays in seeking help.”

LL 494 not only to improve animal welfare but also for a correct management of pets when the owner is not able to take care and to address their needs. We clarified that that’s what animal welfare means here (Line 549) “An emergency care plan for pets is advisable to improve welfare of pets by ensuring they are correctly managed and reduce health-risk to people by reducing delays in seeking help.”

LL 523-525 Sorry but I cannot find this point in the results. This is a qualitative finding described in the Table 9 “Meeting physical needs of dog with 1-walk-a-day rule” and “Becoming bored with same routes/scenery every day”

In general, I suggest to revise the discussion according to the results description. The first results presented were on the walking duration and frequency and also the discussion should be done following the same structure, in this way it is easier reading and understanding.

 The focal point of this study is changing in walking during pandemic but the discussion about this item seems poor and not central. The results of dog walking frequency and duration are discussed in the first and second paragraph of the results, and thus are already primarily highlighted.

Reviewer 2 Report

Thanks for studying important topic for this specific time. 

There are some issues that need to be improved.

Line 43: please specify May 2020 or May 2021

Line 194: please give more details about scoring "energy"

Line 209: please take care of empty spaces (es, n=51)

Line 213-216: it is difficult to follow the sense of the sentence. It would be better to split different results. 

Line 216: Table 1 is in bold in line 216 but not in 218

p value sometimes are in capital letter, other no.

Table 1: check capital letters in first column. What do you mean with "6th form"

Line 222:Cragg-Uhler is not explained in method part

Table 2: "Within-group change- dogs" column , the first number is not explained 

Line 315: double brackets in (Table2)

Table 3: Kruskal-Wallis test-statistic might be abbreviated 

Table 4: it is not clear what you indicate in brackets and not 

Table 9: McNemar’s X2 values present a comma

during lockdown (n=) are not in brackets  

Author Response

Thank you for providing us with constructive comments and suggestions. Below we summarise how these were addressed: 

Comments and Suggestions for Authors

Thanks for studying important topic for this specific time. There are some issues that need to be improved.

Line 43: please specify May 2020 or May 2021- May 2020 has been added (Line 43)

Line 194: please give more details about scoring "energy"- we added a clarification in line (120-121) “dog’s general energy levels as perceived by the respondent (i.e. no further definitions were provided)”

Line 209: please take care of empty spaces (es, n=51)- this was corrected (line 216)

Line 213-216: it is difficult to follow the sense of the sentence. It would be better to split different results. This section now reads: “A multivariable regression model showed that longer walk duration before lockdown was associated with the following dog-related characteristics: large/giant dog size (compared to small/toy size) (P=0.01) and high energy levels (compared to perceived low energy levels) (P<0.001). In the same model, owner-related characteristics associated with longer walk duration included owners aged 30-50 years old (P=0.001) and over 50 years of age (P=0.03) compared to 18-30 years (Table 1). (lines 270-273)

Line 216: Table 1 is in bold in line 216 but not in 218 – amended (line 277)

p value sometimes are in capital letter, other no.- changed throughout the document to capital letter

Table 1: check capital letters in first column. What do you mean with "6th form"- For clarity, this has been re-labelled to “Below A-level or equivalent” throughout the manuscript. Small letters are used instead of capital (Tables 1, 6, Supplementary Material Table S1 and S2).

Line 222:Cragg-Uhler is not explained in method part- a statement “Model fit is reported by describing F value, degrees of freedom, Cragg-Uhler Pseudo R2, and Akaiko Information Criteria (AIC)” was added in line 198-199. This is a standard reporting format recommended by e.g. APA. R2 explains the percentage of variability in response variable explained by the model .  Pseudo  R2 are provided when model is not specified as linear, as in this case (generalised linear model was used instead of linear model).

Table 2: "Within-group change- dogs" column , the first number is not explained “Within-group change (P)” label was added.

Line 315: double brackets in (Table2) removed

Table 3: Kruskal-Wallis test-statistic might be abbreviated – abbreviation introduced in line 216 and applied throughout Tables 3 and 4.

Table 4: it is not clear what you indicate in brackets and not – median and inter-quartile range (IQR) label was now moved from the first column to the second to improve readability. To be consistent, the same change was applied to Table 2.   

Table 9: McNemar’s X2 values present a comma-columns headings were amended to “Pre-lockdown; n, %”

during lockdown (n=) are not in brackets  - column headings amended, so this change is no longer needed.

Reviewer 3 Report

The present study aimed at investigating how the first COVID-19 UK lockdown impacted dog walking practices. Data was collected through an online survey to dog owners in the UK.

The study was carefully designed, the data properly analyzed, the results correctly interpreted and the limitations of the study wisely acknowledged. I am happy to recommend this manuscript for publication after only some minor changes/typos, which I detail in what follows.

Line 130: “Some close-ended questions included ‘Other- please specify’ answers” – Isn’t this an open-ended question? If the respondent is asked to specify?

Line 152: “Data distributions were explored visually and with Shapiro-Wilks tests and visually.” – I believe “visually” is repeated.

Line 295: Parenthesis missing after “medium (P=0.021, Table 3”

Line 315: Please remove one parenthesis in “((Table 2)”.

Line 471: Please remove space after “[17, 19, 37]”

Author Response

Thank you for providing us with constructive comments and suggestions. Below we summarise how these were addressed. 

 Line 130: “Some close-ended questions included ‘Other- please specify’ answers” – Isn’t this an open-ended question? If the respondent is asked to specify? This is still a close-ended question, just with an open text for other option (these were re-coded into pre-existing categories rather than treated in the way open-ended questions were analysed).

Line 152: “Data distributions were explored visually and with Shapiro-Wilks tests and visually.” – I believe “visually” is repeated. This has now been amended (visually at the end of the sentence removed). (Line 202)

Line 295: Parenthesis missing after “medium (P=0.021, Table 3”- added in line 346.

 Line 315: Please remove one parenthesis in “((Table 2)”.removed

 Line 471: Please remove space after “[17, 19, 37]” removed (Line 522)

Other changes made:

Table 5 and Top of Table 7- p-value notation scientific notation converted to real numbers.

Table 7- BIC value removed for consistency of reporting.

Table 2- V-statistic added.

Introduction- small stylistic changes in clarifications (lines 45, 49, 51,66-67, 73,  74, 75, 76, 82

Round 2

Reviewer 1 Report

First of all I would thank the authors for the great job and to have addressed all my comments and clarified my doubts.

This study has permitted to describe the situation during the lockdown on dog walking and ownership in UK and how the lockdown has impacted on these aspects (and viceversa).

The description the questionnaire permitted to understand better the survey design and methodology. 

The introduction of figures and graphs has improved the part of the results. Also the section of "supplementary materials" is very well done and useful in the understanding of the manuscript.

I have no other comments to do.

Author Response

Thank you very much for your comments and further feedback. 

Best wishes

Sara 

Reviewer 2 Report

Dear Authors, 

thanks a lot for your efforts in improving the manuscript. 

I did not understand your change "(i.e. no further definitions were provided)” for defining "dog’s general energy levels". This can create misunderstanding in results and discussion 

Author Response

Thank you for your further comments. 

We have clarified the definition of a dog's energy levels as follows (line 121): 

owner’s subjective assessment of their dog’s perceived energy levels (response categories were: low energy, medium energy and high energy; no further definitions or descriptions of behaviours that exemplify these categories were provided)

We hope this is sufficient clarification and thank you for drawing this to our attention.